# Role of Peroxisome Proliferator-Activated Receptors (PPARs) in Trophoblast Functions

**DOI:** 10.3390/ijms22010433

**Published:** 2021-01-04

**Authors:** Lin Peng, Huixia Yang, Yao Ye, Zhi Ma, Christina Kuhn, Martina Rahmeh, Sven Mahner, Antonis Makrigiannakis, Udo Jeschke, Viktoria von Schönfeldt

**Affiliations:** 1Department of Gynaecology and Obstetrics, Campus Großhadern, Ludwig-Maximilians University of Munich, Marchioninistr. 15, 81377 Munich and Campus Innenstadt: Maistr. 11, 80337 Munich, Germany; Lin.Peng@med.uni-muenchen.de (L.P.); Huixia.yang@med.uni-muenchen.de (H.Y.); Zhi.Ma@med.uni-muenchen.de (Z.M.); martina.rahmeh@med.uni-muenchen.de (M.R.); sven.mahner@med.uni-muenchen.de (S.M.); 2Department of Gynecology and Obstetrics, Zhongshan Hospital, Fu Dan University School of Medicine, Fenglin Rd. 180, Shanghai 200030, China; yeyaoeryida@163.com; 3Center of Gynecological Endocrinology and Reproductive Medicine, Department of Gynecology and Obstetrics, Ludwig-Maximilians University of Munich, Marchioninistr. 15, 81377 Munich, Germany; christina.kuhn@uk-augsburg.de (C.K.); viktoria.schoenfeldt@med.uni-muenchen.de (V.v.S.); 4Department of Gynecology and Obstetrics, Medical School, University of Crete, Andrea Kalokerinou 13, 715 00 Giofirakia, Greece; makrigia@uoc.gr; 5Department of Gynecology and Obstetrics, University Hospital Augsburg, Stenglinstr. 2, 86156 Augsburg, Germany

**Keywords:** peroxisome proliferator-activated receptors (PPARs), cytotrophoblast, extravillous trophoblast, functions

## Abstract

Peroxisome proliferator-activated receptors (PPARα, PPAR*β/δ*, and PPAR*γ*) belong to the transcription factor family, and they are highly expressed in all types of trophoblast during pregnancy. The present review discusses currently published papers that are related to the regulation of PPARs via lipid metabolism, glucose metabolism, and amino acid metabolism to affect trophoblast physiological conditions, including differentiation, maturation, secretion, fusion, proliferation, migration, and invasion. Recent pieces of evidence have proven that the dysfunctions of PPARs in trophoblast lead to several related pregnancy diseases such as recurrent miscarriage, preeclampsia, intrauterine growth restriction, and gestational diabetes mellitus. Moreover, the underlying mechanisms of PPARs in the control of these processes have been discussed as well. Finally, this review’s purposes are to provide more knowledge about the role of PPARs in normal and disturbed pregnancy with trophoblast, so as to find PPAR ligands as a potential therapeutic target in the treatment and prevention of adverse pregnancy outcomes.

## 1. Introduction

Peroxisome proliferator-activated receptors (PPARs) belong to the steroid receptor superfamily, and play roles in both physiological and pathological processes. PPARs regulate cellular biology functions, cell differentiation, metabolic homeostasis, inflammatory response, and immune tolerance [1,2]. PPARs become a therapeutic target for diabetes, metabolic syndrome, cancer, and cardiovascular diseases [3]. Substantial studies have revealed that PPARs play a critical role in the reproduction system [4,5,6]. PPAR isotypes have been found in hypothalamic–pituitary axes and various reproductive tissues, including ovaries, uteri, and placenta in different species [4,6]. The role of PPARs in the regulation of a human’s placental development, endometrium decidualization, and embryo implantation has been demonstrated [7,8,9,10]. In the present review, we summarize recent advances to highlight the possible mediatory role of PPARs and their downstream target genes that directly or indirectly affect trophoblast differentiation, maturation, secretory function, fusion, proliferation, migration, and invasion during pregnancy.

## 2. Biological Functions of PPARs

PPARs, along with the retinoic acid receptors (RARs, RXRs), the thyroid hormone receptors (TRs), the liver X receptors (LXRs), the vitamin D3 receptors (VDRs), and the steroid receptors (SRs), are ligand-activated transcription factors and belong to the nuclear receptor family [11,12]. Three subtypes have so far been described, namely PPARα, PPAR*β*/*δ*, and PPAR*γ*, which are encoded by separate genes labeled PPARA, PPARD, and PPARG, respectively [13,14].

The activation of PPARs requires heterodimerization with the nuclear receptor, RXR. PPARs/RXRs can be activated by 9-cis retinoic acid or natural or synthetic PPAR ligands [15]. The PPAR–RXR complex recruits other cofactors before binding to the PPAR responsive element (PPRE) at the promoter regions of PPAR-responsive genes, usually 5′-AACT AGGNCA A AGGTCA-3′ [16], thus activating or repressing their activities [17].

Each PPAR isoform has tissue-specific expression patterns that are correlated with their distinct functions. PPARα is described as a regulator of fatty acid metabolism [18]. Endogenous ligands include long-chain polyunsaturated fatty acids (LC-PUFAs), such as arachidonic acid, linoleic acid, docosahexaenoic acid, and leukotriene B4 (LTB4) [19]. Synthetic ligands include the fibrates that are pharmacological PPAR*α* agonists used to treat dyslipidemias [20]. PPAR*β*/*δ* is involved in cell differentiation, lipid accumulation, and polarization [21]. Prostaglandin I_2_ (PGI_2_) and diverse PUFAs are natural PPAR*β*/*δ* agonists [22]. Carbaprostacyclin (cPGI_2_) and iloprost are drugs that activate PPAR*β*/*δ* [20]. PPAR*γ* plays a pivotal role in adipogenesis, inflammation, and glucose metabolism [23,24]. PPAR*γ* can be activated by diverse natural ligand PUFAs, including eicosanoids, fatty acids, and oxidized low-density lipoprotein. The bioactive metabolite of prostaglandin D2, 15-Deoxy-12,14-prostaglandin J2 (15dPGJ2), is a potent natural PPAR*γ* agonist [25]. Additionally, rosiglitazone, thiazolidinedione, ciglitazone, troglitazone, and GW1929 are synthetic ligands for PPAR*γ* and involved in insulin-sensitizing activity, fatty acid uptake, and accumulation [26]. The description of agents that activate PPARs is presented in Table 1.

PPARs predominantly localize to the nucleus. Interestingly, PPARs can shuttle between the nucleus and cytoplasm, to regulate biological functions through multiple signals. PPAR*α* and PPAR*γ* nuclear transport is mediated by two nuclear localization signals (NLSs) in DNA-binding domain (DBD)–hinge and activation function 1 (AF1) regions, and their respective receptors include importin α/β, importin 7, and an unidentified receptor. Furthermore, the shuttling of PPAR*γ* from nuclear to cytoplasmic is mediated via PPAR ligands and concentration Ca^2+^ [27]. PPAR*γ* transport from nuclear to cytoplasmic occurs via interaction with the extracellular signal-regulated kinase (ERK) cascade, a component of the mitogen-activated protein kinase (MAPK)/MEK 1/2 [28]. The localization and the mechanism of action of PPARs are shown in Figure 1.

## 3. PPARs in Trophoblast Differentiation

Trophoblast comes from the Greek word “tropho” that refers to feeding the epithelial cell in the placenta. This transient organ plays a pivotal role in fetal growth and development during pregnancy [29]. For a human, the trophectoderm (TE) expands during the early post-implantation period to form a shell around the embryo, thus being composed mostly of self-proliferative cytotrophoblast (CTB). CTB cells act as stem cells with the ability of continuous proliferation and cell differentiation [30]. CTB cells subsequently differentiate into two main subtypes: in the villous pathway, CTBs fuse into syncytiotrophoblast (STB), thereby forming the main site for serving as the gas and nutrient exchange fetal–maternal interface, while others undergo the epithelial-to-mesenchymal transition to differentiate into invasive extravillous trophoblast (EVT) [30,31]. EVT can be further divided into two subtypes: one subtype invades deeply into the uterine wall as groups of cells, termed interstitial EVT, and the other invades the maternal decidual arterioles as endovascular EVT [32] (Figure 2). Trophoblast differentiation is a multifactorial and dynamic process that bridges the exchange between mother and fetus as well as the endocrine functions of the placenta. Three isotypes of PPARs and RXRs are mainly in STB and CTB in the first trimester of the placenta, suggesting that PPARs play specific roles in trophoblast differentiation and functions of the placenta. Data concerning PPARs and RXRs expression are summarized and referenced in Table 2.

In mice, the lack of PPARγ leads to embryonic lethality due to implantation defects. Thanks to MORE-PGKO mice, we understood that PPARγ expression in trophoblasts was sufficient to rescue embryonic lethality and confirmed that PPARγ was necessary for adipogenesis and normal insulin sensitivity [37]. PPARγ stimulates trophoblast differentiation to activate downstream target genes [38,39], and this regulation process is mainly mediated through chorion-specific transcription factor-1 (GCM-1) and the increased expression of chorionic gonadotropin beta-subunit (hCGβ). The loss of the ERK/MAPK cascade caused the reduced expression of PPARγ, affected GCM1 expression, and probably contributed to abnormal CTB differentiation into the outer STB layer and defective STB differentiation, thus leading to the accumulation of multinucleated trophoblast giant cells [40,41,42]. The activation of PPARγ increases hCG secretion in the first- and third-trimester primary villous trophoblast [43,44]. PPARγ/RXRα heterodimers are functional units that increase the transcript levels of hCGβ and the secretion of hCGβ, which can directly stimulate CTB differentiation into the STB [38,45,46]. GCM-1 and hCGβ are regarded as the biochemical markers of trophoblast differentiation [47]. Interestingly, researchers provided evidence between microRNA and PPARγ in the regulation of trophoblast differentiation. MiR-1246 promoted STB differentiation by inhibiting the WNT/β-catenin signaling pathway and increasing the expression of the crucial transcription factors PPARγ and C/EBPβ [48] (Figure 3).

Surprisingly, the role of PPARγ in trophoblast differentiation is controversial, depending mainly on synthetic or natural ligand, trophoblast subpopulation, and gestational age and type. Human primary trophoblast exposure to troglitazone, a synthetic PPARγ ligand, promoted the trophoblast differentiation. However, 15deltaPGJ_2_, a natural PPARγ ligand, hindered the process and promoted the trophoblast apoptosis with the upregulation of P53 [34]. PPARs also exerted an opposite effect on endocrine villous cytotrophoblasts (VCT) and invasion EVT. PPARγ activated the increase of the transcript levels of hCG α- and β-subunit in VCT, whereas they decreased in EVT [49]. Barak Tali et al. suggested that PPARγ is diversely expressed in different trophoblast subsets and times during the trophoblast stem cell differentiation, targeting different genes [50]. The expression of Ldhb and Pcx exhibit PPARγ-independent expression in undifferentiated trophoblast stem cells, but inhibition during early differentiation [50]. Taken together, the data demonstrated that PPARγ has pleiotropic effects capable of changing spatially and temporally, and can affect any subtype of trophoblast lineage differentiation during the pregnancy.

There is also the link among hypoxia, PPARs, and trophoblast differentiation. Daoud et al. proposed that cobalt, a mimic hypoxia chemical agent, inhibited the heart and liver fatty acid-binding protein (H-, L-FABP) and PPAR expression and impaired trophoblast cell differentiation by inducing Hash-2 expression [51]. Hypoxia downregulated the expression of PPARγ in trophoblast, thus decreasing labyrinthine differentiation to trophoblast stem cells by the mechanism independence of both hypoxia-inducible factor (HIF) and histone deacetylases (HDACs) [52]. HIF is known for regulating trophoblast differentiation [53,54,55]. Iodine stimulated the HIF-1α signal to induce oxidative stress with the activation of transcription factor PPARγ, thus leading to abnormal differentiation and migration of trophoblast [56]. Liao et al. put forward that PPARγ and HIF-α are involved in oxidative stress and metabolic stress that are caused by TCDD-induced trophoblast toxicity and the impairment of the placenta vascular network [57]. Hypoxia decreased the expression of TATA-binding protein (TBP-2) and breast cancer resistance protein (BCRP) that are induced by suppressing PPARα and γ, then hampered trophoblast differentiation [58,59]. One of the PPAR ligands, DHA, can stimulate the expression of angiogenesis growth factors, including vascular endothelial growth factor (VEGF), and the PPARβ ligand (GW501516) stimulates another angiogenesis ANGPTL4 expression in first-trimester trophoblast cells, implying that DHA mediates the tube formation process by regulating the secretion of VEGF in first-trimester trophoblast cells [60]. As VEGF and ANGPTL4 expression are increased by hypoxia in the regulation of tight junction structure and function in trophoblast cells [61], the findings correlate PPARs and hypoxia with abnormal trophoblast differentiation and placental insufficiency syndromes of preeclampsia (PE) and intrauterine growth restriction (IUGR) [62,63] (Figure 4).

PPARγ involvement in the process of trophoblast differentiation through hCG and GCM-1 is unquestionable [38]. The PPARα ligand, fenofibrate, could not affect hCG production in human trophoblast in vitro [64]. Accumulating evidence from the literature has shown that PPARβ plays an essential role in embryonic implantation and placentation [65,66], while PPARβ is less understood than PPARγ in human trophoblast differentiation. PPARβ^−/−^trophoblast occurs during default differentiation and invasion [67]. PPARβ promoted the differentiation of trophoblast giant cells, which is associated with lipid metabolism through the activation of the PI3K/Akt signaling pathway and the regulation of the expression of transcription factor I-MFA [68]. It is necessary to continue exploring the function of PPARγ’s participation in the regulation of trophoblast differentiation because its potential as a therapeutic target cannot be ignored.

## 4. PPARs in Other Functions of Trophoblasts

The implantation of the embryo involves a complex process of trophoblast secretion, maturation, fusion, proliferation, migration, and invasion during the first trimester of the placenta.

### 4.1. Secretion

Zhang et al. discovered that the PPARγ signaling pathway is involved in the regulation of visfatin by interleukin (IL)-6 in BeWo, indicating that PPARγ might promote the energy metabolism of trophoblast cells through the secretion of inflammatory cytokines [69]. PPARβ agonist L-165,041 inhibited the synthesis of tumor necrosis factor (TNF)-α, fatty acid-binding protein 3 (FABP3) and IL-6 in porcine trophoblast cells [70]. The PPARγ agonist (rosiglitazone) enhanced the levels of interferon (IFN)-γ and prostaglandin E2 (PGE2) in the incubation medium of trophoblast and mediated by MAPKs [70]. PPARγ seemed to affect the inflammation by interacting with NF-κB in HTR-8/SVneo [71]. The secretion of cytokines is significantly important and extremely complex for implantation during pregnancy, and the functions of PPARs affecting the secretion of trophoblast remain somewhat unknown.

### 4.2. Fusion and Maturation

Following blastocyst adhesion, villous CTBs undergo cell fusion, so as to form the multinuclear STB, which exchanges the nutrients, gas, and metabolites in the fetal–maternal interface [72]. PPARγ/RXRα signaling directly modulates the target gene syncytin-1 through the MAPK or cAMP/PKA pathway, thus contributing to the formation of STB [73]. The PPARβ agonist GW501516 increased the trophoblast cell fusion gene syncytin-A (Syna) but not syncytin-B (Synb), illustrating that PPARβ is correlated with trophoblast cell fusion [74]. PPARγ-deficient mice embryos inhibited the maturation of labyrinthine trilaminar trophoblast and caused defective vascular development [39].

### 4.3. Proliferation, Migration, and Invasion

As a direct transcription target of PPARγ, ANGPTL4 mediated the survival, proliferation, migration, and invasion of HTR-8/SVneo in vitro [75]. The PPARγ agonist, pioglitazone, could promote EVT migration via stimulating insulin-like growth factor (IGF) signaling, which is induced by improving insulin sensitivity [76]. The activation of PPARγ by PPARγ synthetic and natural ligands inhibited HIPEC 65 cell line invasion without affecting proliferation [35,43,77,78]. Furthermore, the molecular mechanism of PPARγ might modulate trophoblast invasion by decreasing pregnancy-associated plasma protein-A (PAPP-A) and inhibiting the secretion of insulin-like growth factor (IFGII) [79]. In addition, other researchers have suggested that lysyl oxidase (LOX), LOXL1, and LOXL2 are negative regulators of PPARγ downstream targets that affect trophoblast invasion [80]. Matrix metalloproteinase (MMP)-2 and MMP-9 were also reported as downstream genes of PPARγ, thus inhibiting the trophoblast invasion [81]. The effects of PPARγ impact on trophoblast, depending on different model trophoblast cell types. Meanwhile, it has displayed different sensitivities even though with the same thiazolidinedione concentration. Indeed, the different PPARγ ligands might affect PPARγ at its transcriptional or post-transcriptional levels, interfering with normal trophoblast invasion. Inadequate trophoblast invasion is thought to result in preeclampsia and recurrent miscarriage. Our group’s results are consistent with other findings, showing that PPARγ downregulated the expression in the trophoblast of recurrent miscarriage [82,83]. Further research studies have demonstrated that PPARγ played an essential role in the mono ethylhexyl phthalate (MEHP)-inhibited trophoblast invasion by disturbing the balance of MMP-9 and tissue inhibitors of metalloproteinase (TIMP)-1 expression in early pregnancy loss [84]. Rosiglitazone, as PPARγ agonist, applies the reduced uterine perfusion pressure (RUPP) rat model of PE via a heme oxygenase 1-dependent pathway [85].

## 5. PPARs and Energy Metabolism in the Trophoblast

Nutrients transported from the maternal to fetal circulation cross the villous trophoblast. Among nutrients, lipid from lipoproteins, glucose, and amino acids are essential for energy supply, steroid hormone synthesis, and fetus growth. The maternal profile influences placenta FA uptake through human trophoblasts [86].

PPARs modulate fat transport, fat storage, and fat metabolism in trophoblast. The lipid droplet that is associated with protein adipophilin existed in human trophoblast and was enhanced during trophoblast differentiation and upregulated by PPARγ/RXR [87]. The level of adipophilin is upregulated by troglitazone to activate PPARγ in trophoblast based on microarray experiments [88]. The underlying mechanism of PPARγ/RXR enhanced the neutral lipids and the uptake of the free fatty acids, and the expression of fatty acid transport protein 4 (FATP4) might be activated by p38 MAPK in trophoblasts [89,90]. They also observed that RXR activation could decrease FATP2 in trophoblast [89]. In contrast to former bodies of evidence, MARK4 promoted lipid accumulation by activating WNT/β-catenin and inhibiting PPARγ in the pig trophoblast [91]. Schild et al. reported that oxidized lipids such as 9S-hydroxy-10E,12Z-octadecadienoic acid (9-HODE), 13S-hydroxy-9Z,11E-octadecadienoic acid (13-HODE), and 15S-hydroxy-5Z,8Z,11Z,13E-eicosatetraenoic acid (15-HETE) could also stimulate PPARγ activity by increasing hCG in cultured term human trophoblasts [92].

The PPARγ agonists thiazolidinedione and pioglitazone induced the expression of visfatin by IL-6, implying that PPARγ promotes trophoblasts’ energy metabolism [69]. Hyperglycemia activates PPARγ pathways, thus decreasing the invasion of human CTB, followed by the increase of IL-6 and soluble fms-like tyrosine kinase-1 (sFIt-1) and the inhibition of urokinase plasminogen activator (uPA) and plasminogen activator inhibitor 1 (PAI-1) [93]. Recent evidence from Cawyer et al. proved that hyperglycemia induced human CTB apoptosis and antiangiogenesis with the upregulation of PPARγ and p38 MAPK phosphorylation [94], thus confirming the role of PPARγ in glucose metabolism that induced CTB dysfunction. The ability of PPARγ to regulate glucose homeostasis requires further investigation in multiple mechanisms.

PPARs are linked with insulin signaling and amino acid transport in trophoblast. The activation of PPARα inhibits IRS-1 and AKT phosphorylation, thus leading insulin to stimulate the expression of SNAT2 and amino acid transport in trophoblast, which is essential for fetal growth [95]. PPARγ was necessary for adipogenesis and normal insulin sensitivity; therefore, presenting PPARγ expression in trophoblast could rescue embryonic lethality [37]. Insulin sensitizer could increase the expression of PPARγ in primary extra villous trophoblast [76]. Adiponectin hampered insulin-stimulated amino acid uptake in cultured primary human trophoblast cells by modulating insulin receptor substrate phosphorylation [96].

These findings pointed out that the master regulator PPARγ/RXR, which dramatically alters fat, glucose and amino acid metabolism in trophoblasts, might provide clues to cure pregnancy metabolic diseases such as gestational diabetes mellitus (GDM) and PE.

## 6. Conclusions

In summary, our review summarizes the current bodies of evidence about the role of PPARs in trophoblast differentiation, including secretion, fusion, maturation, proliferation, migration, and invasion, which is associated with energy metabolism. The dysregulation of these human trophoblast physiology processes can cause gestational diseases, including recurrent miscarriage, GDM, PE, and IUGR. PPAR ligands, especially rosiglitazone and 15dPGJ2, should be further investigated as well as their potential as therapeutic therapy for abnormal trophoblast function in related gestational diseases.

## Figures and Tables

**Figure 1 ijms-22-00433-f001:**
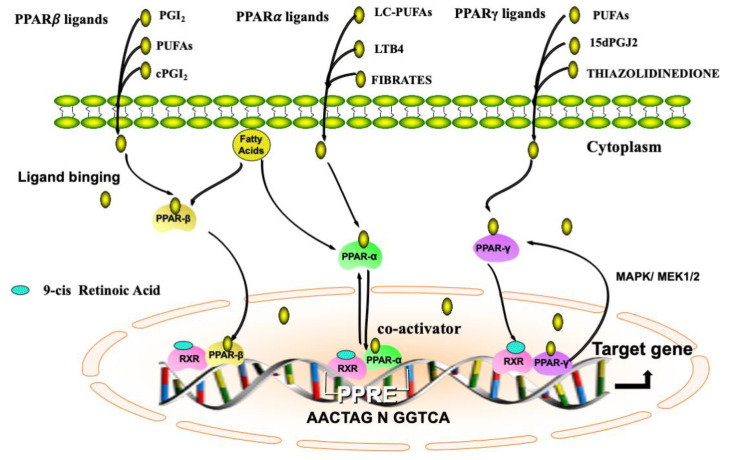
Illustrative representation of localization and mechanism of action of PPARs. PPARs belong to the nuclear receptor superfamily and consist of PPARα, PPAR*β*/*δ*, and PPAR*γ*. Endogenous PPAR ligands can be transferred from the cytosol or generated in the nuclear membrane. PPARs bind to the specific DNA sequence AACT AGGNCA A AGGTCA, PPAR responsive element (PPRE), and their activation requires heterodimerization with another nuclear receptor, RXR. PPARs modulate numerous target genes by co-activator or co-inhibitor activity. PPARs are mainly expressed in the nucleus, and could shuttle from nuclear to cytoplasm. PPAR*γ* transport from nuclear to cytoplasmic occurs via the extracellular signal-regulated kinase (ERK) cascade, a component of the mitogen-activated protein kinase (MAPK)/MEK 1/2. PPARs, peroxisome proliferator-activated receptors; RXR, 9-cis retinoic acid receptor.

**Figure 2 ijms-22-00433-f002:**
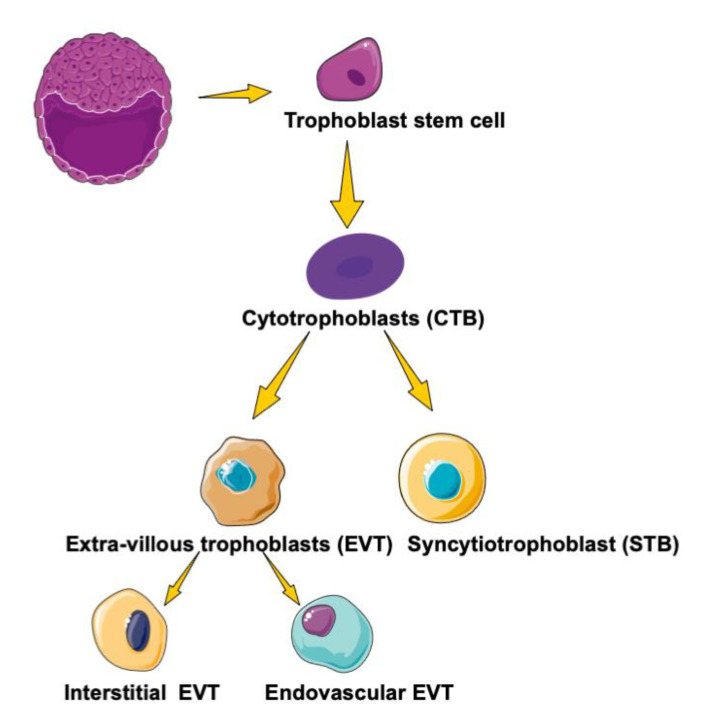
The path of human trophoblast differentiation. Trophoblast stem cells are derived from the trophectoderm. The cytotrophoblasts differentiate in two ways: invasive extravillous trophoblast (EVT) and syncytiotrophoblast (STB). EVT can be divided into two subtypes, one of which invades deeply into the uterine wall as groups of cells, termed interstitial EVT, and the other invades the maternal decidual arterioles as endovascular EVT.

**Figure 3 ijms-22-00433-f003:**
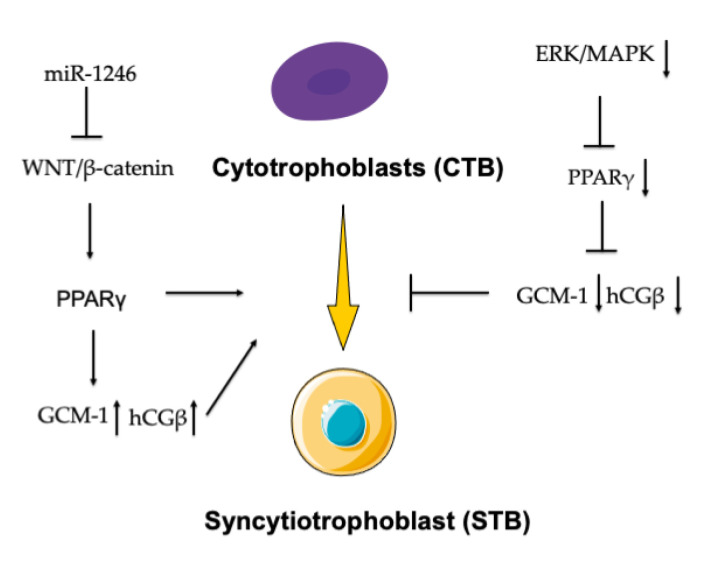
PPARγ involved in trophoblast differentiation. The activation of PPARγ increased the expression of chorionic gonadotropin beta-subunit (hCGβ) and chorion-specific transcription factor (GCM-1), which promoted cytotrophoblast (CTB) differentiation into the STB. miR-1246 promoted STB differentiation through inhibiting the WNT/β-catenin signaling pathway and increasing the expression of the crucial transcription factor PPARγ. The loss of the ERK/MAPK cascade caused the reduced expression of PPARγ, affected the expression of GCM1, and inhibited the process of CTB differentiation into the STB.

**Figure 4 ijms-22-00433-f004:**
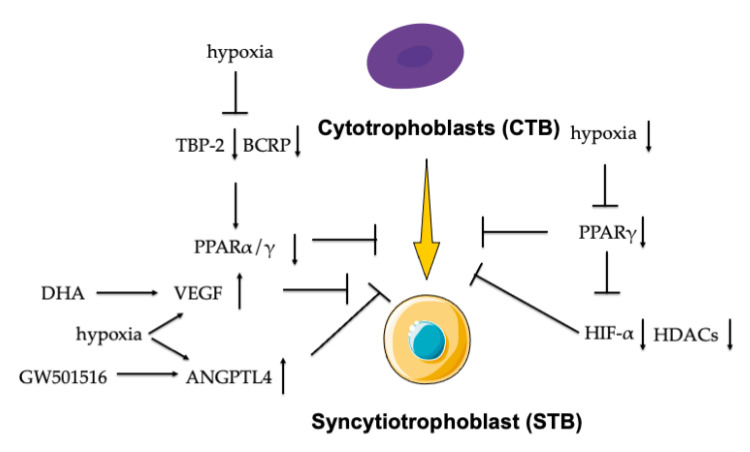
PPARs linked hypoxia with trophoblast differentiation. The hypoxia inhibited the expression of PPARγ with the inactivation of both hypoxia inducible factor (HIF) and histone deacetylases (HDACs), which hampered the trophoblast differentiation. Hypoxia decreased the expression of TBP-2 and breast cancer resistance protein (BCRP) induced by suppressing PPARα and γ, which affected trophoblast differentiation. DHA increased the secretion of VEGF and the PPARβ ligand (GW501516) stimulated ANGPTL4 expression, which influenced the trophoblast differentiation.

**Table 1 ijms-22-00433-t001:** Natural and synthetic ligands of peroxisome proliferator-activated receptors (PPARs).

Receptor	Natural Ligands	Synthetic Ligands
PPARα	arachidonic acid	fibrates
	linoleic acid	industrial plasticizers
	leukotriene B4	naveglitazar
	docosahexaenoic acid	netoglitazone
	eicosapentaenoic acid	muraglitazar
	prostaglandin I_2_ (PGI_2_)	
PPAR*β*/*δ*	polyunsaturated fatty acids (PUFAs)	carbaprostacyclin (cPGI_2_)
	eicosanoids	iloprost
PPAR*γ*	fatty acids	rosiglitazone
	oxidized low-density lipoprotein	thiazolidinedione
	15-deoxy-12,14 prostaglandin J2 (15dPGJ2),	ciglitazone
	prostaglandin D2	troglitazone
	13- hydroxyoctadecadienoic acid (HODE)	GW1929

**Table 2 ijms-22-00433-t002:** Expression of PPARs and RXRs in human trophoblasts.

	Location	Methodology	References
PPARα	STB, CTB	RT-PCR, IHC	[33]
PPARβ	STB, CTB	RT-PCR, IHC	[33]
PPARγ	STB, CTB, EVT	RT-PCR, IHC, Microarray	[33,34,35,36]
RXRα	STB, CTB, EVT	RT-PCR, IHC	[33,35]
RXRβ	Not detected	RT-PCR, IHC	[33]
RXRγ	STB, CTB	RT-PCR, IHC	[33]

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
