# Peer review of "Role of Peroxisome Proliferator-Activated Receptors (PPARs) in Trophoblast Functions"

_ijms, 2021, doi:10.3390/ijms22010433_

Round 1

Reviewer 1 Report

The manuscript has been changed substantially. However, there are some mistakes or incomprehensible sentences left. I have listed them below.

L. 34: delete „they”, it should be “and play roles”

L. 38: it should be “have been found” or “were found”

L. 40: delete “involved”, it should be “The role of PPARs in the regulation”

L. 43: change “secretion” into “secretory function”, otherwise it sounds as trophoblast is secreted, what does not make sense

L. 46: delete “including”, it should be “along with” or “beside”; as a result “PPARs along with the retinoic …”

L. 48: delete “a member”. It should be “belong to the nuclear receptor family”

L. 54-55: it should be “region”; then, delete “PPRE generally … genes” and extend the previous sentence as follows: “… [16], thus activating or repressing gene transcription [17]” to avoid unnecessary repeats.

L. 57: Begin with “Each PPAR isoform…”

L. 66: eicosanoids are not PUFAs

L. 67: replace “include” with “of” to make “The bioactive metabolite of prostaglandin…”

L. 70-71: Maybe change into: “The description of agents which activate PPARs is presented in Table 1”

L. 73: remove “so as”

L. 80: it should be “shows Figure 1”

L. 90: maybe change into: “… superfamily and consists of “, not “composed”

L. 93: “modulate” nor “are modulated” and “PPARs are mainly expressed in nucleus…”

L. 97: it should be “PPARS in trophoblast differentiation”, not “trophoblasts”

L. 100-101: this statement is controversial, remove it. Trophoblast consist of trophectoderm and endoderm cells.

L. 102-103: this sentences is incomprehensible.

L. 107: it should be “one” not “One”

L. 112: delete “may”, it should be “PPARs play specific…”

L. 113: maybe change into “Data concerning PPAR and RXR expression are summarized and referenced in Table 2”.

L. 123: Begin with “In mice, the lack of…”

L. 125: it should be “PPARg stimulates”

L. 126: maybe change into “.. and this regulation process..”

L. 145: I suggest to change “mainly depending” into “and the obtained results depend mainly on”

L. 147: it should be “exposure to”

L. 152-153: this sentence is incomprehensible

L. 155: it should be “this data” not “dates”

L. 166: change “migration with” into “migration of”

L. 170-171: maybe “One of PPAR ligand, DHA…” or “One of PUFAs, DHA …”

L. 215-216: incomprehensible sentence; maybe “PPARg-deficient mouse embryos showed inhibited maturation of labyrinthines trilaminar trophoblast and defects in vascular ??? [39]”. Something is missing at the end of this sentence, did Author mean “vascular development”?

L. 219: Begin with “PPARg agonist, pioglitazone…”

L. 223: It should be “Furthermore, the molecular…”

L. 228-229: incomprehensible sentence, improve it

L. 234: di Authors mean “recurrent miscarriages”? delete “group”

L. 241: change the tile into “The involvement of PPARs in energy metabolism…” or “PPARs and energy metabolism in …”

L. 268-269: delete “Researchers displayed that”, begin with “The activation…”

L. 280: maybe change into “… evidences about the role of PPARs in trophoblast…”

Author Response

Reviewer #1:

  1. Response to comment: L. 34: delete „they”, it should be “and play roles”

 Response: We have made correction according to your comments.

  1. Response to comment: L. 38: it should be “have been found” or “were found”

 Response: We have made correction as your suggestion.

  1. Response to comment: L. 40: delete “involved”, it should be “The role of PPARs in the regulation”

 Response: We have made correction as your suggestion.

  1. Response to comment: change “secretion” into “secretory function”, otherwise it sounds as trophoblast is secreted, what does not make sense

 Response: We have made correction according to your comments.

  1. Response to comment: L. 46: delete “including”, it should be “along with” or “beside”; as a result “PPARs along with the retinoic …”

 Response: We have made correction according to your comments.

  1. Response to comment: L. 48: delete “a member”. It should be “belong to the nuclear receptor family”

 Response: We have deleted the “a member” according to your comments.

  1. Response to comment: L. 54-55: it should be “region”; then, delete “PPRE generally … genes” and extend the previous sentence as follows: “… [16], thus activating or repressing gene transcription [17]” to avoid unnecessary repeats.

 Response: It is really true as your suggested that the former write seemed repeat.

  1. Response to comment: L. 57: Begin with “Each PPAR isoform…”

 Response: We have revised the part as your suggested.

  1. Response to comment: L. 66: eicosanoids are not PUFAs

 Response: We have deleted the eicosanoids acid in the revised manuscript.

  1. Response to comment: L. 67: replace “include” with “of” to make “The bioactive metabolite of prostaglandin…”

 Response: We have made correction according to your comments.

  1. Response to comment: L. 70-71: Maybe change into: “The description of agents which activate PPARs is presented in Table 1”

 Response: It is really true as your suggested that this description is clearer.

  1. Response to comment: L. 73: remove “so as”

 Response: We have deleted so as in the revised manuscript.

13.Response to comment: L. 80: it should be “shows Figure 1”

 Response: We have changed the word to shows according to your comments.

14.Response to comment: L. 90: maybe change into: “… superfamily and consists of “, not “composed”

 Response: We have replaced composed with consists of as your suggestion.

15.Response to comment: L. 93: “modulate” nor “are modulated” and “PPARs are mainly expressed in nucleus…”

 Response: We have changed the word to shows according to your comments.

16.Response to comment: L. 97: it should be “PPARS in trophoblast differentiation”, not “trophoblasts”

Response: We have changed the word to shows according to your comments.

  1. Response to comment: L. 100-101: this statement is controversial, remove it. Trophoblast consist of trophectoderm and endoderm cells.

Response: We have removed the sentence according to your comments.

  1. Response to comment: L. 102-103: this sentences is incomprehensible.

Response: We have rewritten this sentence that CTB cells act as stem cell with the ability of continuous proliferation and cell differentiation.

  1. Response to comment: L. 107: it should be “one” not “One”

Response: We have made correction according to your comment.

  1. Response to comment: L. 112: delete “may”, it should be “PPARs play specific…”

Response: We have made correction according to your suggestion.

  1. Response to comment: L. 113: maybe change into “Data concerning PPAR and RXR expression are summarized and referenced in Table 2”.

Response: We have made correction according to your comment.

  1. Response to comment: L. 123: Begin with “In mice, the lack of…”

Response: We have made correction according to your suggestion.

  1. Response to comment: L. 125: it should be “PPARg stimulates”

Response: We have made correction according to your comment.

  1. Response to comment: L. 126: maybe change into “.. and this regulation process..”

Response: We have made correction according to your suggestion.

  1. Response to comment: L. 145: I suggest to change “mainly depending” into “and the obtained results depend mainly on”

Response: We have made correction according to your suggestion.

  1. Response to comment: L. 147: it should be “exposure to”

Response: We have made correction according to your comment.

  1. Response to comment: L. 152-153: this sentence is incomprehensible

Response: We have rewritten this sentence that PPARγ diversely express in different trophoblast subsets and times during the trophoblast stem cells differentiation with targeting different genes.

  1. Response to comment: L. 155: it should be “this data” not “dates”

Response: We have made correction according to your suggestion.

  1. Response to comment: L. 166: change “migration with” into “migration of”

Response: We have changed the word “migration with” into “migration of”

  1. Response to comment: L. 170-171: maybe “One of PPAR ligand, DHA…” or “One of PUFAs, DHA …”

Response: We have changed into One of PPAR ligand, DHA…

  1. Response to comment: L. 215-216: incomprehensible sentence; maybe “PPARg-deficient mouse embryos showed inhibited maturation of labyrinthines trilaminar trophoblast and defects in vascular ??? [39]”. Something is missing at the end of this sentence, did Author mean “vascular development”?

Response: We are very sorry for our negligence of the word development.

  1. Response to comment: L. 219: Begin with “PPARg agonist, pioglitazone…”

Response: We have made correction according to your comment.

  1. Response to comment: L. 223: It should be “Furthermore, the molecular…”

Response: We have made correction according to your comment.

  1. Response to comment: L. 228-229: incomprehensible sentence, improve it

Response: We have re-written this part according to your suggestion that the effects of PPARγ impact on trophoblast depend on different model trophoblast cell types. Meanwhile, it displayed different sensitivities even though with the same thiazolidinedione concentration.

  1. Response to comment: L. 234: di Authors mean “recurrent miscarriages”? delete “group”

Response: We have deleted the word group.

  1. Response to comment: L. 241: change the tile into “The involvement of PPARs in energy metabolism…” or “PPARs and energy metabolism in …”

Response: We have changed the subtitle into PPARs and Energy Metabolism in the Trophoblast.

  1. Response to comment: L. 268-269: delete “Researchers displayed that”, begin with “The activation…”

Response: We have made correction according to your comment.

  1. Response to comment: L. 280: maybe change into “… evidences about the role of PPARs in trophoblast…”

Response: We have made correction as your suggestion.

Special thanks to you for your good and detailed comments.

Reviewer 2 Report

Line 123 “It has been reported that as for mice, the lack of PPARγ leads to embryonic lethality due to severe lipodystrophy, hypotension and insulin resistance. The effects can be rescued by preserving PPARγ in the trophoblast [37].”

Please, rewrite this sentence, in this form is confusing. It has been reported that the lack of PPARγ leads to embryonic lethality due to implantation defects. Thanks to MORE-PGKO mice, we understand that PPARγ expression in trophoblasts was sufficient to rescue embryonic lethality and confirmed that PPARγ was necessary for adipogenesis and normal insulin sensitivity.

Line 138 curial??????

Line 149, please change PPARs  with PPARgamma

Line 166, please change PPARs  with PPARgamma

Minor

Line 185 “With the role of PPARγ in the process of trophoblast differentiation in unquestionable, PPARα ligand, fenofibrate, did not affect hCG production in human trophoblast in vitro” Please correct the sentence

Line 268, please change PPARs is with PPArs are

Line 215 Please correct vascular

Line 223 “Furthermore, The molecular……” Please correct

Author Response

Reviewer #2:

  1. Response to comment: Line 123 “It has been reported that as for mice, the lack of PPARγleads to embryonic lethality due to severe lipodystrophy, hypotension and insulin resistance. The effects can be rescued by preserving PPARγin the trophoblast [37].”

Please, rewrite this sentence, in this form is confusing. It has been reported that the lack of PPARγ leads to embryonic lethality due to implantation defects. Thanks to MORE-PGKO mice, we understand that PPARγexpression in trophoblasts was sufficient to rescue embryonic lethality and confirmed that PPARγ was necessary for adipogenesis and normal insulin sensitivity.

Response: It is really clearer as your suggested that in mice, the lack of PPARγ leads to embryonic lethality due to implantation defects. Thanks to MORE-PGKO mice, we understood that PPARγ expression in trophoblasts was sufficient to rescue embryonic lethality and confirmed that PPARγ was necessary for adipogenesis and normal insulin sensitivity.

  1. Response to comment: Line 138 curial??????

Response: We are very sorry for our incorrect writing and corrected the “curial “to “crucial “.

3.Response to comment: Line 149, please change PPARs with PPARgamma

Response: We have made correction as your suggestion.

4.Response to comment: Line 166, please change PPARs with PPARgamma

Response: We have made correction as your suggestion.

5.Response to comment: Line 185 “With the role of PPARγ in the process of trophoblast differentiation in unquestionable, PPARαligand, fenofibrate, did not affect hCG production in human trophoblast in vitro” Please correct the sentence

Response: PPARγ involvement in the process of trophoblast differentiation through hCG and GCM-1 is unquestionable[38]. PPARα ligand, fenofibrate, could not affect hCG production in human trophoblast in vitro[64].

6.Response to comment: Line 268, please change PPARs is with PPArs are

Response: We have corrected “is” into “are”.

7.Response to comment: Line 215 Please correct vascular

Response: We have corrected into vascular development.

8.Response to comment: Line 223 “Furthermore, The molecular……” Please correct

Response: We have changed the word “The” into “the”.

This manuscript is a resubmission of an earlier submission. The following is a list of the peer review reports and author responses from that submission.

Round 1

Reviewer 1 Report

The manuscript “Role of Peroxisome proliferator-activated receptors  (PPARs) in reproduction and pregnancy” could be interesting because there are a lot of studies on this field, and reviews, that in critical way, summarize these study are extremely necessary . However the Authors do not discuss the data with a critical view, rather the manuscript looks like a list of information; the figures are not appropriately done and the references result wrong or absent. Please the Authors have to check the manuscript also for many typos and for acronyms.

  1.  

Major: line 54, the ref.8 is not correct! Please check all the reference! The reference especially in a review manuscript are very important for the readers! Moreover the Authors have to pay attention on PPARs ligands, for example the iloprost activity on pparb/d is controversial.

Overall the panel of PPAR’s agonists is not complete, lack the discussion regarding  the  PPARα/γ/δ pan-agonists or PPARα/γ dual agonist. I think a use of a description table is necessary!

Minor: page 2 line 5 “….polarization”, I think the Authors are speaking about keratinocytes polarization! Fig1  refers to PPARs mechanism of action not structure, so delete line 68…”structure of ” in the legend. Moreover the Authors put figure 1 in the main text to describe the PPAR’s cellular function, it is not true; please correct!

  1. Physiological role of PPARs in pregnancy

PPARs in HPG axis

Major: The Authors should be critical on this paragraph, for example in the Line 92 -the discrepancies between different species-, they have to explain it! Moreover the Authors, speaking about the role of PPARs in pituitary function, continue, without a logical link, to discuss about an in vitro study on ovarian follicles. Finally, the figure does not clarify the roles of PPARs in HPG axis.

I think this paragraph needs of a deep revision.

Minor: line 87 PPARg it's written in italics

PPARs in Ovary

Major: Perhaps the ovarian tissue should be described by specifying the expression of the PPARs. Recent work has highlighted, for example, a different expression of PPARs and RXRs in human granulosa cells. Moreover, It should be specified on which kind of animals the experiments were carried out. The effect of PPARs on ovarian function is completely missing, it should be added, perhaps in the form of a table.

Overall when present the Authors have to cite the works on human.

PPARs in Uterus

Major: The authors, for an immediate understanding of the manuscript, should describe on which kind of animal the experiments were conducted.

PPARs in Trophoblast Function of Placenta

Major: The authors again should describe on which kind of animal the experiments were conducted.

Minor: Format to line 166 to 180

  1. PPARs Implications in Reproductive Diseases

PPARs in Unexplained Recurrent Pregnancy Losses

Major: again the Authors do a list of information without a critical discussion, perhaps the link between PPARs and endocrine system should be cited (some ligands of the PPARs are endocrine disruptors); the link between the PPARs and lifestyle should be discussed too.

Line 228 “…for heart development in placenta.”???? What does it mean?

Line235 the Ref.80 is an in vitro work, I think the sentence “in early pregnancy loss” is a speculation.

Line 319: Several studies disclosed that the use of PPARγ agonists have numerous side effects, the Authors have to discuss this!

polycystic ovary syndrome (PCOS)

PCOS should be explained in more detail; there's a lot of bibliography about it.

The Authors have to mentioned that in PCOS patients were reported a PPARγ polymorphism (Pro12Ala) and many other works on PPARg and PCOS.

Overall, the manuscript is suitable for publication only after an extensive revision.

Author Response

Thank you very much for your evaluation.

Because the former review was lacking the "leading idea", we wrote a new review about the:

"Role of Peroxisome proliferator-activated receptors (PPARs) in trophoblast functions"

Because the review is rewritten in total we ask for a new evaluation of this manuscript. 

Reviewer 2 Report

Major concerns:

  1. The manuscript lacks “leading idea”, it is not clear what scientific story did Authors plan to describe. It seems that there was no solid plan for this review, because no new data or theory concerning regulation of expression or role of PPARs were provided. Most of data, information, conclusions included in the manuscript could be easily found in other already published reviews. I understand that this is a review, so published results are described, but it should serve to describe some new ideas, hypotheses, or suggestions.
  2. Section 1 and 2 are quite messed up. Try to organize these parts by dedicating each paragraph to one isoform, or one tissue, or one biological process. Cited results and information are disordered a lot in the present manuscript. It is hardly to follow given information, understand and remember them. Several sentences are taken out of context. Moreover, accompanied by sentences like “The ovary produces estrogen and progesterone”, that is quite obvious information. Section 2 gives the impression that Authors are not closely familiar with regulations of HPG axis function and endometrium development/receptivity in general.
  3. The manuscript needs substantial English revision; there are so many grammar and phrasal errors, syntax errors, inappropriate construction of sentences, what makes them incomprehensible. Some sentences suddenly end, like those in lines 196-197 or 230-231.
  4. Avoid too many “Some researchers have reported”, “they also indicated”, “study demonstrated”, or “other data showed”. Give your own comment to results you describe! There must be a clear reason that this paper has been written. And the Reader should feel that.
  5. Authors cite some results about PPARs without giving the name of the tissue, or organ, or species these results concern. The Reader feels confused a lot.

Specific comments:

Title: pregnancy is a part of reproduction, very important one; so, why to separate pregnancy from reproduction? Moreover, half of the manuscript is dedicated to various reproductive complications, but not mentioned in the title.

Abstract: to be honest, PPARs are not really “highly” expressed in reproductive tissue; their role is predominately associated with lipid and cholesterol metabolism and glucose homeostasis. Their expression and role in reproductive tract is in fact marginal compared to their expression and role in adipose tissue, heart, liver, or kidney.

Lines 87-90: Authors wrote that rosiglitazone modulates pituitary function directly, but then they describe an indirect action of this PPARgamma agonist.

Figure 2 is incomprehensible, not clear what the Authors demonstrate here; all these interactions between hypothalamus, pituitary and ovary are completely not clear for the Reader; legend to Fig. 2 does not help to understand these. Gonads do not look like gonads at all.

Lines 105-106: This sentence contain one of several in this manuscript strange formulations, which do not sound scientific; “The ovary function is …secreting female gametes”? Hormones may be secreted, proteins, steroids, but not gametes. Another one “oocyte maturation of the ovary” in line 116. And many, many more.

Lines 270-277: This paragraph does not explain RIF.

Figure 3. Schematic presentation of the uterus dominates over the data provided for each tissue/organ. Moreover, this is a uterus of a woman, while several results described in this figure concern mice and pigs, which possess uteri of a completely different shape, structure.

Gene nomenclature throughout the whole manuscript should follow a standard guidelines provided by HUGO Gene Nomenclature Committee. There are differences in writing gene names between human, rodents, and other species. Moreover, abbreviation for protein is not always the same as abbreviation for gene or mRNA.

Author Response

(The authors gave the same response as above.)
